# An Evaluation of Antarctic Sea-Ice Thickness from the Global Ice-Ocean Modeling and Assimilation System based on In situ and Satellite Observations

Sutao Liao[1], Hao Luo[1*], Jinfei Wang[1], Qian Shi[1], Jinlun Zhang[2], Qinghua Yang[1]

[1]School of Atmospheric Sciences, Sun Yat-sen University, and Southern Marine Science and Engineering Guangdong Laboratory (Zhuhai), Zhuhai, 519082, China

[2]Polar Science Center, Applied Physics Lab, University of Washington, Seattle, WA 98105

*Correspondence to*: Hao Luo (luohao25@mail.sysu.edu.cn)

**Abstract.** Antarctic sea ice is an important component of the Earth system. However, its role in the Earth system is still unclear due to limited Antarctic sea-ice thickness (SIT) data. A reliable sea-ice reanalysis can be useful to study Antarctic SIT and its role in the Earth system. Among various Antarctic sea-ice reanalyses products, the Global Ice-Ocean Modeling and Assimilation System (GIOMAS) output is widely used in the researches of Antarctic sea ice. As more Antarctic SIT observations with quality control are released, a further evaluation of Antarctic SIT from GIOMAS is conducted in this study based on in situ and satellite observations. Generally, though only sea-ice concentration is assimilated, GIOMAS can basically reproduce the observed variability of sea-ice volume and its changes in the trend before and after 2013, indicating that GIOMAS is a good option to study the long-term variation of Antarctic sea ice. However, due to deficiencies in the model and asymmetric changes in SIT caused by assimilation, GIOMAS underestimates Antarctic SIT especially in deformed ice regions, which has an impact on not only the mean state of SIT but also the variability. Thus, besides the further development of the model, assimilating additional sea-ice observations (e.g., SIT and sea-ice drift) with advanced assimilation methods may be conducive to a more accurate estimation of Antarctic SIT.

## 1    Introduction

Antarctic sea ice plays an important role in the Earth system. Firstly, Antarctic sea ice can influence the Earth climate system. For instance, changes in Antarctic sea ice could affect freshwater flux of the

Southern Ocean that directly influences the stratification of the ocean (Goosse and Zunz, 2014; Haumann
et al., 2016). Besides, Antarctic sea ice acts as a protective buffer for Antarctic ice shelves, with the
thinning or absence of sea ice increasing the possibility of ice shelf disintegration (Robel, 2017; Massom
et al., 2018). Secondly, Antarctic sea ice has a significant impact on the biosphere of the Earth system.
Studies have shown that the variation of Antarctic sea-ice thickness (SIT) will affect the maximum
biomass of algae in different ice layers, which will influence the food web of the Southern Ocean
(Massom and Stammerjohn, 2010; Schultz, 2013). Thirdly, Antarctic sea ice has impacts on human
activities such as shipping and fishery management (Dahood et al., 2019; Mishra et al., 2021). Hence,
studies on Antarctic sea ice are of great scientific and socio-economic importance.
To truly understand changes in sea ice of the Southern Ocean, SIT is needed to estimate the sea-ice
volume (SIV), since it is through volume changes that sea ice has its greatest impact on the water column
(Maksym et al., 2012; Hobbs et al., 2016). Although changes in Antarctic sea-ice extent (SIE) have been
investigated extensively (Turner et al., 2015; Parkinson, 2019), they may not be a robust proxy of large-
scale changes in SIV as there are differences between the variation of SIV and SIE in some regions of
the Antarctic (e.g., Kurtz and Markus, 2012). Many studies related to Antarctic sea ice are limited by the
lack of reliable SIT data. For example, freshwater flux of the Southern Ocean, which affects the
stratification of ocean, cannot be accurately estimated as part of the freshwater flux comes from sea-ice
melting and growth (Haumann et al., 2016). In addition, the skill of sea-ice prediction cannot meet the
need of human activities in the Antarctic (Mishra et al., 2021). Studies have shown that the skill of
Antarctic sea-ice prediction could be improved with better SIT initialization (Bushuk et al., 2021). So
far, the commonly used types of the Antarctic SIT data are observations, model data, and reanalyses
products and each type of data has its own limitations.
Antarctic SIT observations can be divided into in situ and satellite observations. In situ observations can
provide the local state of Antarctic SIT. However, the sparse distribution of in situ SIT observations pose
considerable challenges to understand the large-scale characteristics of SIT (Worby et al., 2008a). It is
well known that satellite observations have wider spatiotemporal coverage than in situ observations.
However, previous studies indicate that there is large uncertainty in SIT data retrieval from satellite
altimeters owing to the relatively small freeboard (i.e., thickness of sea ice or sea ice and snow above the
sea surface) of Antarctic sea ice compared to that in the Arctic (Maksym and Markus, 2008) and the lack
of knowledge about coincident snow cover thickness as well as sea ice and snow density (Alexandrov et

al., 2010). In addition, results of numerical simulations are used to investigate the long-term variation of Antarctic SIT (Zhang, 2007; Holland et al., 2014), but discrepancies are identified not only between models and observations but also among models (Shu et al., 2015; Tsujino et al., 2020), indicating the large uncertainty in model estimates.

It should be noted that reanalyses have unique advantages over observed and simulated SIT. Theoretically, reanalyses can provide more accurate or comprehensive state estimations than can otherwise be obtained through either observations or models alone (Buehner et al., 2017). Reanalyses merges the information from both observations and models through data assimilation. Compared with observations, reanalyses data can provide coordinated and gridded data with homogenous sampling in time and space over a long period (Parker, 2016). Besides, compared with model only data, reanalyses data can produce the state estimations closer to observations because of data assimilation (Lindsay and Zhang, 2006; Rollenhagen et al., 2009). Hence, SIT reanalyses have been widely adopted in studies on the Antarctic sea ice (Abernathey et al., 2016; Kumar et al., 2017). Nevertheless, there are still large uncertainties of present sea-ice reanalyses in the Southern Ocean (Uotila et al., 2019; Shi et al., 2021), suggesting the necessity and importance of evaluating them.

Among a number of Antarctic sea-ice reanalyses, the Global Ice-Ocean Modeling and Assimilation System (GIOMAS) is one of the most widely used in studies of Antarctic sea ice. For instance, GIOMAS has been regarded as the reference in the assessments of simulations (Shu et al., 2015; Uotila et al., 2017; DuVivier et al., 2020) and predictions (Ordoñez et al., 2018; Morioka et al., 2021). However, GIOMAS has been less widely evaluated, in part because there are far fewer observations of Antarctic SIT against which evaluation is possible (DuVivier et al., 2020).

Due to advances in observing technology as well as algorithms in recent years, the quality of Antarctic SIT observations is improved. For example, compared to the European Remote-Sensing Satellites (i.e., ERS-1 and ERS-2), the Synthetic-Aperture Interferometric Radar Altimeter (SIRAL) on board CryoSat-2 (CS2) is equipped with two radar antennas, which significantly improves the accuracy of sea-ice freeboard (i.e., thickness of sea ice above the sea surface). CS2 also has a much wider spatial coverage with improved along-track resolution because of the design of the satellite orbit and multiple operation modes (Parrinello et al., 2018). In addition, Paul et al. (2018) developed an adaptive retracker threshold for CS2 to produce a consistent sea-ice freeboard data. Besides, more Antarctic SIT observations has been available with the accumulation of observations. For instance, more in situ observations are

obtained from dedicated research stations, icebreakers and autonomous underwater vehicles due to
increasing research activities in the Antarctic. These provide an opportunity to further evaluate Antarctic
SIT of GIOMAS. Notably, since in situ observations provide relatively accurate estimations in specific
points while satellite data provides relatively long and continuous observations with wide spatial
coverage, various observations are adopted in the evaluation to make it more comprehensive.
The paper is organized as follows. In Sect. 2, Antarctic SIT from GIOMAS and observations are
introduced. In Sect. 3, Antarctic SIT of GIOMAS is evaluated with observations from different aspects,
including the climatology, the linear trend, the intensity of variability, as well as the frequency
distribution. The final section provides the conclusions and discussion.
**2    Data and methods**
**2.1    SIT from GIOMAS**
GIOMAS consists of a global Parallel Ocean and sea Ice Model (POIM) with data assimilation
capabilities, which is developed at University of Washington (Zhang and Rothrock, 2003). The ocean
component of POIM is the Parallel Ocean Program, and the sea-ice component of POIM is the 8-category
thickness and enthalpy distribution sea-ice model. The National Centres for Environmental Prediction-
National Centre for Atmospheric Research (NCEP-NCAR) daily reanalysis (Kalnay et al., 1996)
provides the atmospheric forcing for POIM. Furthermore, in GIOMAS, the modelled sea-ice
concentration (SIC) is nudged towards observed SIC derived from Special Sensor Microwave Imager
launched by the Defense Meteorological Satellite Program (Weaver et al., 1987), and other modelled
variables including SIT are adjusted subsequently. The detailed adjustment process of SIT is as follows:
when SIC is nudged in the system, it will modify the SIT distribution to accommodate the change in SIC,
which remove sea ice from the distribution without considering its SIT if modelled SIC is too large, while
add sea ice to the 0.1-m ice thickness bin if modelled SIC is too small. (Lindsay and Zhang, 2006). This
process can reduce the root-mean-square difference and improve the correlation between modelled SIT
and observed SIT and will also cause the thinning of the mean SIT. More technical details for POIM and
assimilation procedures can be found in Zhang and Rothrock (2003) and Lindsay and Zhang (2006),
respectively. GIOMAS data is available from 1979 to the present with a global coverage and data
involved in the assessment spans from January 1979 to December 2018 (Fig. 1a). The average horizontal
spatial resolution is 0.8 degrees of longitude × 0.8 degrees of latitude (around 60 km × 60 km), and the
temporal resolution is one month for all variables. Additionally, GIOMAS also provides daily outputs
for some variables including SIT, SIC and snow depth, and the daily SIT of GIOMAS is assessed in this
study. SIT data of GIOMAS is the equivalent SIT, which represents SIV per unit area.
**2.2    SIT from satellite altimeters and in situ observations**
Satellite-altimeter observations involved in this study are from radar altimeters on board Envisat (ES)
and CS2, which are generated by the Sea Ice Climate Change Initiative (SICCI) project under the
European Space Agency Climate Change Initiative (ESA CCI) program. ES was equipped with the Radar
Altimeter 2, measuring sea-ice freeboard mainly based on $K_u$-band frequency (Hendricks et al., 2018b).
The Antarctic SIT data derived from ES freeboard spans from December 2002 to November 2011 (Fig.
1a) with a coverage of the entire Antarctica (Fig. 1b). The spatial resolution is 50 km × 50 km and the
temporal resolution is 1 month. CS2 was equipped with the SIRAL, measuring the sea-ice freeboard
mainly based on $K_u$-band frequency like ES (Hendricks et al., 2018a). CS2 Antarctic SIT dataset spans
from November 2010 to April 2017 (Fig. 1a) and the spatial coverage and the spatiotemporal resolution
are the same as ES SIT dataset.
In situ SIT observations involved in this study are from upward looking sonar (ULS), ship-based and air-
based measurements. ULS is a kind of mooring measurement at fixed locations, measuring sea-ice draft
(thickness of sea ice below the water surface) with a time interval shorter than 15 minutes (Behrendt et
al., 2013). Ice draft needs to be converted into total SIT in an empirical way according to Harms et al.
(2001). Thirteen ULSs used in this study were deployed in the Weddell Sea (Fig. 1b) by Alfred Wegener
Institute (AWI) and spanned from 1990 to 2010 intermittently (Fig. 1a).
The ship-based observations are made up of the Antarctic Sea Ice Processes & Climate program
(ASPeCt), ANT-XXIX/6 (Schwegmann, 2013) and ANT-XXIX/7 (Ricker, 2016). The ASPeCt dataset
not only includes ASPeCt observations collected from 1981 to 2005 (Worby et al., 2008a), but also
includes the ASPeCt bridge-based sea-ice observations collected from 2007 to 2012. The ship tracks
cover all sectors of the Southern Ocean (Fig. 1b) and the average spacing of data points is 6 nautical
miles. The air-based SIT observations include data collected by the air-based electromagnetic system
(i.e., like an electromagnetic bird carried by the helicopter) with a high frequency of 0.5 Hz and an

average spacing of 3-4 m (Lemke, 2009, 2014), which is mainly located in the northwest Weddell Sea (Fig. 1b).

Since both satellite observations and in situ observations are involved in the evaluation, it is necessary to investigate the relationship between them. To achieve this, direct comparisons among satellite observations and ULS observations are conducted in the Weddell Sea where ULS SIT observations are available. The monthly SIT of ES and CS2 during their coincident segments (i.e., November 2010 to November 2011) in the Weddell Sea is displayed in Fig. 2a. The SIT of ES and CS2 is mainly distributed around the one-to-one line and there is a significant correlation of 0.69 between them, indicating ES-derived SIT is comparable to CS2-derived SIT. Then the monthly ES SIT is compared with monthly ULS SIT during the coincident segments at sites 206, 207, 208, 229, 231 and 233 (Fig. 2b). CS2 dataset is not involved since there are only four data pairs between CS2 and ULS observations. The distribution of data pairs indicates ES tends to overestimate SIT compared with ULS observations, because the scattering surface of the radar altimeter can be inside the snow (Willatt et al., 2010; Wang et al., 2020). However, 77% of ULS SIT is within the uncertainty of ES (Fig. 2b). Besides, the correlation between the ULS SIT that is within the uncertainty of ES and the corresponding ES SIT is 0.73. All those indicate ES-derived SIT is comparable to ULS observations when the uncertainty is considered.

## 2.3    Data processing and methods

According to Parkinson and Cavalieri (2012), the summer, autumn, winter and spring refer to January-March, April-June, July-September and October-December, respectively. As shown in Fig. 2b, the Southern Ocean is divided into the Weddell Sea (60° W-20° E), the Indian Ocean (20° E-90°E), the western Pacific Ocean (90°E-160°E), the Ross Sea (160°E-130°W) and the Amundsen/Bellingshausen Sea (130°W-60°W).

Since the mismatch in spatial and temporal resolutions between reanalyses and observations could introduce substantial representation errors in the comparisons, the data is processed as Janjić et al. (2018) suggested to eliminate such mismatch between GIOMAS and observations. In general, GIOMAS data is converted to the locations of the observations when compared with satellite and ULS observations while the ship-based and air-based observations are converted to gridded data based on GIOMAS grid. For details, when compared with satellite observations, daily GIOMAS data is interpolated to the grid of

satellite observations using the linear approach and converted to monthly averages. For the comparisons between GIOMAS and ULS observations, 15-minutely ULS data is converted to daily averages for comparison with daily GIOMAS data and the nearest neighbour approach is used to find the GIOMAS grid cells closest to the ULS locations. Besides, when compared with ship-based and air-based observations, since the observed data is very dense in space and the temporal resolution is always within one day, it is averaged into daily and gridded data based on the GIOMAS grid to create a proper dataset that is compatible with daily GIOMAS SIT data.

The climatological annual cycle is defined as the multi-year averages in each month. For observations, the climatological annual cycles are calculated from all years available in each observation dataset. For GIOMAS, when compared with satellite observations, GIOMAS data that coincides with the time spans of satellite observations are selected (2002-2011 for ES and 2010-2017 for CS2) to calculate the climatology. When compared with ULS observations, all years available in GIOMAS (1979-2018) are used for the computation of climatology. Anomalies are defined as departures from the climatological annual cycle, and the intensity of variability is defined as the standard deviation of anomalies. During the overlapping time of ES and CS2 (November 2010 to November 2011), though the difference in SIV anomalies between ES and CS2 (i.e., root-mean-square error is 473.1 $km^3$) is not small compared with the mean standard deviation of SIV anomalies (i.e., their standard deviations in ES and CS2 are 960.7 $km^3$ and 956.6 $km^3$, respectively), the selection of data in the coincident segment has little effect on the trend. Thus, the SIV anomalies of CS2 during the overlapping time are chosen and ES from December 2002 to October 2010 and CS2 from November 2010 to April 2017 are combined to obtain a relatively long and continuous SIV time series for the trend computation. In addition, since the trajectories of air-based SIT observations are mainly distributed in the northwest Weddell Sea where is dominated by deformed sea ice (Fig. 2b), the comparison between GIOMAS and air-based observations is only conducted in the Weddell Sea.

## 3    Results

### 3.1    Comparison in the climatology of SIV and SIT

Figure 3 shows the climatological annual cycle of Antarctic SIV. Although obvious uncertainties of SIV can be found in both ES and CS2, the annual cycle of ES is similar to that of CS2. Both ES and CS2

show that the melt rate of sea ice is nearly twice the growth rate. Besides, there are also some differences
in the SIV climatology between ES and CS2. The SIV of CS2 is greater than that of ES in the winter and
spring, and larger uncertainties of SIV can be found in ES. The SIV difference between ES and CS2 may
be owing to the mismatch in the sea-ice freeboard between ES and CS2. As Paul et al. (2018) indicated,
due to the unresolved physical processes such as complex snow metamorphism or sea-ice surface
roughness influenced by the flooding in the snow/ice interface, the sea-ice freeboard of ES cannot be
well matched with the ones of CS2 in the Antarctic though the retracker algorithms are the same.
GIOMAS can reproduce the asymmetry in the annual cycle of Antarctic SIV observed by ES and CS2
while underestimates SIV by about 38% on average when compared to ES as well as CS2. Meanwhile,
the underestimation is seasonally dependent, with weaker underestimation in summer and stronger one
in winter.
Figure 4 shows the spatial distribution of SIT bias in summer as well as winter to investigate details of
SIV underestimation in these two seasons. In both seasons, significant negative SIT bias of GIOMAS
can be found in the deformed ice zone, such as the northwestern Weddell Sea and coasts of the
Amundsen/Bellingshausen Sea as well as the coast of East Antarctic. Meanwhile, the extent of negative
bias is wider in the winter (Figs. 4b and d) rather than in the summer (Figs. 4a and c), which results in
seasonal differences of the SIV underestimation (Fig. 3). In addition, there are weakly positive SIT biases
in the southwestern Weddell Sea during winter (Figs. 4b and d), which may be due to model bias in
simulating sea-ice transport in the western Weddell Sea (Shi et al., 2021). Considering sea-ice
deformation is also related to sea-ice motion tightly, a better simulation of sea-ice motion is required to
achieve more accurate reconstruction of Antarctic SIT. In addition, the relatively large positive bias in
winter Ross Sea SIT can only be found in the comparison between GIOMAS and CS2, which may be
caused by smaller freeboard of CS2 than ES in the winter Ross Sea as shown in Paul et al. (2018). Notably,
some of the radar altimeter signals would originate from the snow/air interface or from somewhere inside
the snow and result in an overestimation of ice freeboard (Willatt et al., 2010; Wang et al., 2020).
These uncertainties, combined with often thick snow and complex snow metamorphism in the Antarctic,
can contribute to the overestimation of the Antarctic SIT from ES and CS2. Thus, the underestimation of
SIT from GIOMAS can be partially attributed to the uncertainties of SIT retrieved from ES and CS2.
However, the underestimation in the deformed ice regions can be owing to the deficiency of GIOMAS
since the differences of SIT between GIOMAS and satellite observations in those regions are always
larger than the uncertainties of satellite observations. To prove this, a direct inter-comparison between
the monthly SIT of GIOMAS, ES and ULS at site 206 is conducted and the biases of GIOMAS and ES
relative to ULS at site 206 and the uncertainties of each dataset are displayed in Table 1. The bias of
GIOMAS is larger than the uncertainty of ULS while the bias of ES is smaller than the uncertainty of
ULS (Tabel 1). This suggests there is a significant discrepancy between GIOMAS and ULS SIT while
ES SIT is comparable to ULS SIT at site 206.
Due to large uncertainties in above satellite observations, the SIT of GIOMAS is further assessed by ULS
measurement in the Weddell Sea. Considering significant variation of sea ice over horizontal distances
as small as a few meters, the standard deviation of ULS is displayed in Fig. 5a. It is obvious that the
variability of ULS near the shore (i.e., 206, 207, 212, 217, 232 and 233) is stronger than that of ULS far
from the shore (i.e., 208, 209, 210, 227, 229, 230 and 231), indicating larger sea-ice deformation near
the shore. As Fig. 5b shows, GIOMAS significantly underestimates the nearshore SIT all year round
while slightly overestimates SIT far from the shore in the winter, implying the deficiency of GIOMAS
in the simulation of sea-ice deformation, which leads to underestimation of SIT in the Weddell Sea from
a perspective of regional average. The above deficiency of GIOMAS might be attributed to the
insufficient resolutions of the model and assimilated SIC observations, which are not able to well resolve
the coastal lines and hinder GIOMAS from reproducing the ice deformation near shore. Therefore,
GIOMAS is indeed to underestimate the climatology of Antarctic SIT, mainly in the deformed sea-ice
zone, compared with satellite and in situ observations. In addition to the model drawbacks of GIOMAS,
this underestimation might also be introduced by the assimilation procedure of GIOMAS. Although only
satellite SIC is nudged in GIOMAS, SIT would be adjusted asymmetrically as described in Sect 2.1. This
asymmetric addition and removal of ice leads to a thinning of the mean ice thickness (Lindsay and Zhang,
2006). Notably, though the uncertainty of satellite observations is large, the differences between
GIOMAS and satellite SIT cannot be ignored since the uncertainty of satellite observations is expected
to be large owing to the difficulties with the estimation of snow depth and density in the Antarctic (Ozsoy-
Cicek et al., 2011; Bunzel et al., 2018).
**3.2    Comparison in the trend of SIV**
Antarctic SIE shows different trends before and after 2014 (Parkinson, 2019). Though there can be a
significant correlation between Antarctic SIE and SIV, differences between the variation of SIV and SIE
cannot be ignored in some regions. Therefore, it is necessary to examine whether there are similar
changes in the trend of Antarctic SIV. As Figure 6 shows, the observed Antarctic SIV anomaly increased
gradually from 2003, reached the maximum (2783 $km^3$) in November 2013, and then abruptly declined
from September 2013 to April 2017. The evolution of SIV anomaly is comparable to that of SIE anomaly,
while the time of the SIV anomaly peak is earlier than that of the SIE anomaly peak nearly by one year.
The trends of GIOMAS and observed SIV anomalies are 989 and 2968 $km^3$ per month before 2013 and
-84762 and -119875 $km^3$ per month after 2013. Although there are differences in the SIV trend between
GIOMAS and satellite observation, GIOMAS can basically reproduce the changes in the observed SIV
trend before and after 2013. Besides, the correlation of SIV anomalies between GIOMAS and
observations is 0.83, which passes a two-tailed t test at 99% significant level. Given the advantages of
reanalyses over observations or models individually especially in the polar region (Buehner et al., 2017),
GIOMAS data would be a good choice to study the variability and long-term trends of Antarctic sea ice.
**3.3    Comparison in the intensity of SIT variability**
Figure 7 displays spatial differences in the intensity of SIT anomalies variability between GIOMAS and
satellite observations. Compared with ES and CS2, GIOMAS underestimates the intensity of SIT
variability in the Southern Ocean, especially in the deformed ice zone (Figs. 7a-b), which resembles the
spatial pattern of Fig. 4. The underestimation in the deformed ice regions can be found also in the
comparison between GIOMAS and ULS. The intensity of SIT variability is underestimated near the shore,
while overestimated away from the shore (Fig. 7c). The spatial distribution of differences in the intensity
of variability is roughly consistent with that of SIT differences in Fig. 5b. These phenomena suggest that
there appears to be a relationship between the mean SIT and the variability. As Blanchard-Wrigglesworth
and Bitz (2014) suggested, models with a thinner mean ice state tend to have SIT anomalies with smaller
amplitude. In addition, the comparison in SIT standard deviation ratio and mean bias between GIOMAS
and satellite observations shown in supplementary figure further clarifies the relationship that with a
negative SIT bias, GIOMAS always underestimates the variability of SIT. Thus, the bias of SIT has an
impact not only on the climatology of SIT but also on the variability of SIT. It should be mentioned that
in the regions where the uncertainty of satellite observations is larger than the difference between
GIOMAS and satellite observations (i.e., mainly in the regions with undeformed sea ice), the uncertainty
would have an impact on the evaluation in the variability of SIT and cannot be ignored.

## 3.4    Comparison of SIT frequency


In addition to ULS observations, the rest of in situ sea-ice observations are sparse in the Southern Ocean
and mainly provided by ship-based and air-based measurements. Figure 8 displays the SIT frequency
distribution of GIOMAS and ship-based as well as air-based in situ observations. The peaks of
observations are mainly around 0-0.6 m while the frequencies of GIOMAS SIT are mainly distributed in
0-1.4 m in the Southern Ocean (Fig. 8a). In different sectors (Figs. 8b-f), the frequency distribution of
observed SIT data is similar to that in the whole Southern Ocean while the peaks of GIOMAS SIT
frequency vary from 0.2 m to 1.4 m. Compared with observations, for the Southern Ocean, GIOMAS
has a higher frequency within 0.6-1.6 m while a lower frequency in the rest bins compared with ship-
based observations (Fig. 8a), which seems to imply the overestimation of SIT. The similar results can be
found in different sectors (Figs. 8b-f). However, the sample selection bias should be noted in the ship-
based observations due to ship's track avoiding areas of thicker ice, which results in its estimation biased
toward thinner ice (Timmermann, 2004; Williams et al., 2015). Besides, GIOMAS has a lower frequency
of thick ice in the Weddell Sea compared with air-based observations. In conclusion, GIOMAS tends to
overestimate SIT frequency between 0.6-1.6 m in the Southern Ocean compared with ship-based
observations under the premise that ship-based observations always bias low. Additionally, the
comparison between GIOMAS and air-based SIT observations further proves the weakness of GIOMAS
in the simulation of sea-ice deformation.

## 4    Conclusions and discussion


Considering the important role of SIT in studies of Antarctic sea ice and the wide application of GIOMAS,
the Antarctic SIT of GIOMAS is assessed with satellite and in situ observations. In general, GIOMAS
can basically reproduce the observed variability and linear trends of SIV even though only satellite SIC
data is assimilated by nudging. For the climatology, GIOMAS can reproduce the asymmetry in the annual
cycle of Antarctic SIV. For the long-term SIV variation, the variation of GIOMAS is in phase with that
of observations, and it is also able to capture the changes in linear trends before and after 2013. These
suggest that GIOMAS is useful to study the long-term variation of Antarctic sea ice. However, significant
negative bias in SIT can be found in the comparison between of GIOMAS and observations. Compared
with satellite measurements, GIOMAS tends to underestimate SIT, especially in regions with strong ice
deformation. This underestimation is of seasonal dependence with greater underestimation in the winter.
Although above underestimation can be partially attributed to the uncertainties of SIT retrieved from
satellite especially in the undeformed ice zone, the differences cannot be ignored and SIT
underestimation in the northwest Weddell Sea is further verified by the comparison between GIOMAS
and ULS observations. Furthermore, the spatial distribution of the differences in the magnitude of SIT
variability resembles that of the differences in SIT climatology between GIOMAS and observations.
Given the relationship between mean state of SIT and variability (Blanchard-Wrigglesworth and Bitz,
2014; also verified by the comparison between satellite observations and GIOMAS in the supplement),
this phenomenon indicates that SIT underestimation might have an impact on not only the SIT
climatology but also the SIT variability. In addition, GIOMAS overestimates SIT compared with ship-
based observations, which can be due to the negative bias in ship-based SIT estimation (Timmermann,
2004; Williams et al., 2015). The deficiency of GIOMAS in simulating deformed sea ice is further
verified in the comparison with air-based observations.
Notably, though GIOMAS could basically reproduce the trends of Antarctic SIV anomalies before and
after 2013, the differences in the trends of SIV anomalies between GIOMAS and satellite observations
cannot be ignored. A simple comparison between the monthly GIOMAS sea-surface temperature (SST)
and Microwave Optimally Interpolated SST observations reveals that the positive bias of GIOMAS in
SST before 2014 is roughly corresponding to the underestimation of positive trend of observed SIV
anomalies while the negative SST bias of GIOMAS after 2014 is corresponding to the underestimation
of negative trend of observed SIV anomalies. There seems to be a possible relationship between the
difference in SST and the difference in the trends of SIV anomalies between GIOMAS and observations
since higher SST would slow down the increase of SIV while lower SST would slow down the decrease
of SIV. However, this relationship needs further quantification and further analysis is added to our future
work plan.
In addition, limitations from Antarctic SIT observations are non-negligible in this study. For one aspect,
the scarcity of Antarctic SIT observations is one of the main sources of limitations for the evaluation.
The time span of satellite observations is not long enough for the evaluation of GIOMAS SIT data from
1979 to the present while the in situ observations are too few to show the estimation of SIT in the entire
Southern Ocean. Those make it unable to comprehensively evaluate the entire GIOMAS Antarctic SIT
data in this study. For another, this study is also limited by observations of Antarctic SIT due to their
unsuitability for the evaluation. For example, though SIT from ICESat (Kern et al., 2016) equipped with
the Geoscience Laser Altimeter System is available from 2004 to 2008 and proved to have lower bias in
SIT estimation than radar altimeter measurements (Willatt et al., 2010; Wang et al., 2020), it is not
adopted in this study. The reasons are as follows. First, ICESat SIT is not available in winter (July-
September), when greater underestimation of SIT is found in GIOMAS (Fig. 3). Second, the data size of
ICESat is relatively smaller than that of ES and CS2 because ICESat provides seasonal mean data and
its time range is narrower. Therefore, the additional assessment on SIT of GIOMAS will be conducted
when the Antarctic SIT derived from ICESat-2 is available. Furthermore, the uncertainty of satellite
observations has an impact on the evaluation and the accuracy of satellite observations needs to be further
improved to obtain more accurate satellite-derived SIT estimations with smaller uncertainty. The
uncertainty of satellite-derived SIT observations is mainly from the uncertainty introduced by the
scattering surface of radar signals and the estimation of Antarctic snow depth and density. With the
influences of complex snow stratigraphy and flooding inside the snow related to the formation of snow
ice, the assumption that the radar signal reflects from the snow/ice interface is not applicable in most
cases (Willatt et al., 2010). Besides, owing to the lack of knowledge of Antarctic snow, the climatology
of snow depth from the European Space Agency-SICCI Advanced Microwave Scanning Radiometer for
the Earth Observing System (AMSR-E) and the Advanced Microwave Scanning Radiometer 2 (AMSR2)
is used in the retrieval of ES and CS2-derived SIT, which would introduce extra uncertainties since the
inter-annual variability in snow depth is omitted (Bunzel et al., 2018). Moreover, the AMSR-E/AMSR2
snow depth is indicated to considerably underestimate the actual snow depth, which usually occurs in
the East Antarctic (Worby et al., 2008b; Ozsoy-Cicek et al., 2011). All those contribute to the large
uncertainty of the satellite-derived SIT in the Antarctic and the uncertainty would influence the
evaluation of SIT in the regions where the differences between GIOMAS SIT and satellite observations
are smaller than the uncertainty. Therefore, a more accurate estimation of Antarctic snow depth and
density would be essential to reducing the uncertainty of satellite SIT observations and thus improving
the reliability of the evaluation.
The above SIT underestimation of GIOMAS can be partially attributed to the model weakness. For
example, insufficient resolution of the model restricts GIOMAS to reproduce the ice deformation near
shore. Besides, the assimilation is a vital component in the reanalyses since it could constraint the model
with observations and make the model obtain better state estimation (Lahoz and Schneider, 2014).
However, it can also be a source of errors in the system. In GIOMAS, the asymmetric SIT changes
introduced by assimilation cannot be ignored. Thus, besides the further development of the model, there
are potential ways to improve the estimation of Antarctic SIT from the perspective of data assimilation.
Firstly, additional sea-ice observations other than SIC should be assimilated. For example, besides
Antarctic SIT derived from Envisat and CryoSat-2 used in this study, the Antarctic SIT retrieved from
ICESat-2 is also to be released in the near future, and hence assimilating these SIT observations directly
may suppress the bias of SIT (e.g., Yang et al., 2014; Fritzner et al., 2019; Luo et al., 2021). Also,
assimilating sea-ice drift observations can improve the simulation of sea-ice motion and deformation,
which can improve the estimation of SIT (e.g., Lindsay and Zhang, 2006; Mu et al., 2020). Secondly,
advanced data assimilation methods should be adopted to provide balanced estimation of model state.
For instance, the innovation of SIC can be converted to the increment of SIT in a more balanced way
through the flow-dependent covariance of Ensemble Kalman Filter (e.g., Massonnet et al., 2013; Yang
et al., 2015). Thirdly, due to the characteristics of SIC observations derived from the passive microwave
instrument, observation errors of SIC should vary with time and location (Lindsay and Zhang, 2006).
However, a fixed value of observation errors is adopted in GIOMAS because of limited information on
observation errors of SIC. Thus, a better estimation of SIC observation errors might further improve the
performance of GIOMAS. Furthermore, though nudging of SIC is not state of the art, it makes the model
of GIOMAS obtain better SIT simulation while the model only data of GIOMAS is likely to overestimate
SIT in the marginal seas. To promote the development of GIOMAS, further quantitative analyses on the
impact of nudging SIC on the SIT in the Antarctic are worthy of attention and will be conducted in the
future.
Besides, in the course of global warming, Antarctic SIE rose gradually and reached a record high in
2014/2015 before decreasing dramatically, which is obviously different from the dramatic drop in Arctic
SIE during the satellite era (e.g., Turner and Comiso, 2017). Results from a recent study suggest that the
trend in Antarctic ice coverage may be due to changes in atmospheric (e.g., Holland and Kwok, 2012)
and oceanic (e.g., Meehl et al., 2019) processes. Without better SIT and SIV estimates, it is difficult to
characterize how Antarctic sea-ice cover is responding to changing climate, or which climate parameters
are most influential (Vaughan et al., 2013). Thus, more Antarctic sea-ice observations and more studies
on data assimilation are urgently needed to accurately evaluate the Antarctic SIT, which can help to
improve the reconstruction and prediction of Antarctic SIV and to support research related to Antarctic

405 sea ice.

## Data availability

The GIOMAS reanalysis data are available at http://psc.apl.washington.edu/zhang/Global_seaice/data.html. The satellite-based Antarctic sea ice thickness observations from Envisat and CryoSat-2 are available at https://doi.org/10.5285/b1f1ac03077b4aa784c5a413a2210bf5 and https://doi.org/10.5285/48fc3d1e8ada405c8486ada522dae9e8, respectively. The Weddell Sea upward-looking sonar sea ice draft data are available at https://doi.pangaea.de/10.1594/PANGAEA.785565. The ship-based sea ice thickness observations are available at http://aspect.antarctica.gov.au/data, https://doi.org/10.1594/PANGAEA.819540 and https://doi.org/10.1594/PANGAEA.831976. The sea ice thickness observations from airborne electromagnetic system are freely available at https://doi.pangaea.de/10.1594/PANGAEA.771229 and https://epic.awi.de/id/eprint/36245/.

## Author contribution

QY and HL developed the concept of the paper. SL and HL performed analysis and drafted the manuscript. JW collected the remote sensing and observation data. QY, JZ, QS and JW gave comments and helped revise the manuscript. All of the coauthors contributed to scientific interpretations.

## Competing interests

The authors declare that they have no conflict of interest.

## Acknowledgments

This study is supported by the National Natural Science Foundation of China (No. 41941009, 41922044, 42006191), and the Guangdong Basic and Applied Basic Research Foundation (No. 2020B1515020025). This is a contribution to the Year of Polar Prediction (YOPP), a flagship activity of the Polar Prediction Project (PPP), initiated by the World Weather Research Programme (WWRP) of the World Meteorological Organisation (WMO). We acknowledge the WMO WWRP for its role in coordinating this international research activity.

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

**Table 1.** The biases of GIOMAS and ES SIT relative to that of ULS and the uncertainties of SIT at ULS
206 (Unit: m).

| Dataset | Bias | Uncertainty |
| --- | --- | --- |
| ULS | 0 | 1.17 |
| ES | 0.89 | 1.37 |
| GIOMAS | -1.99 | 0.34 |



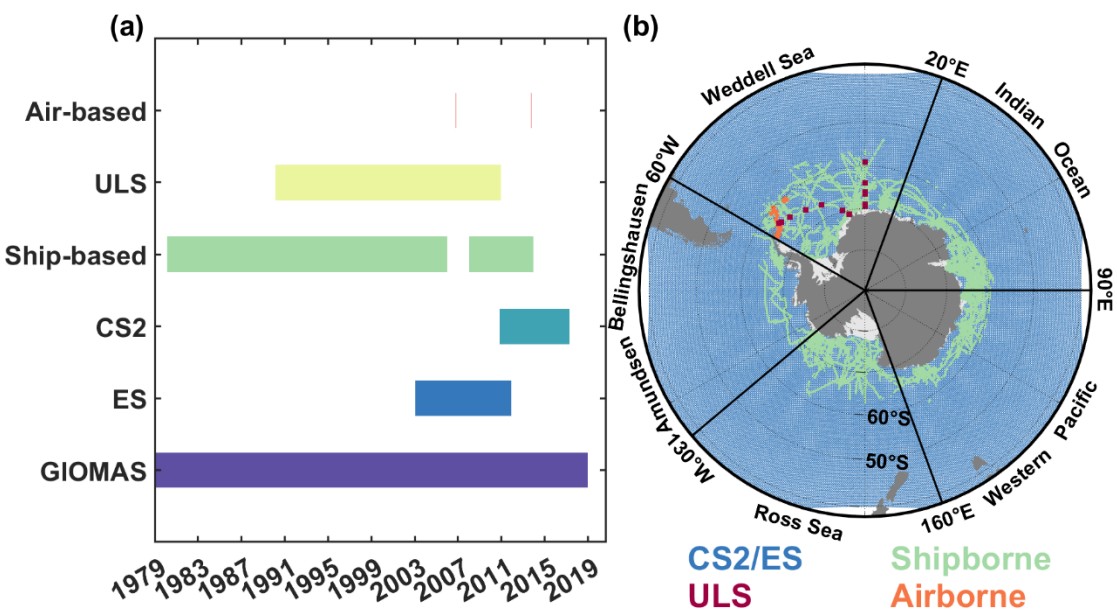


**Figure 1.** (a) The temporal and (b) spatial coverage of data used in this study.

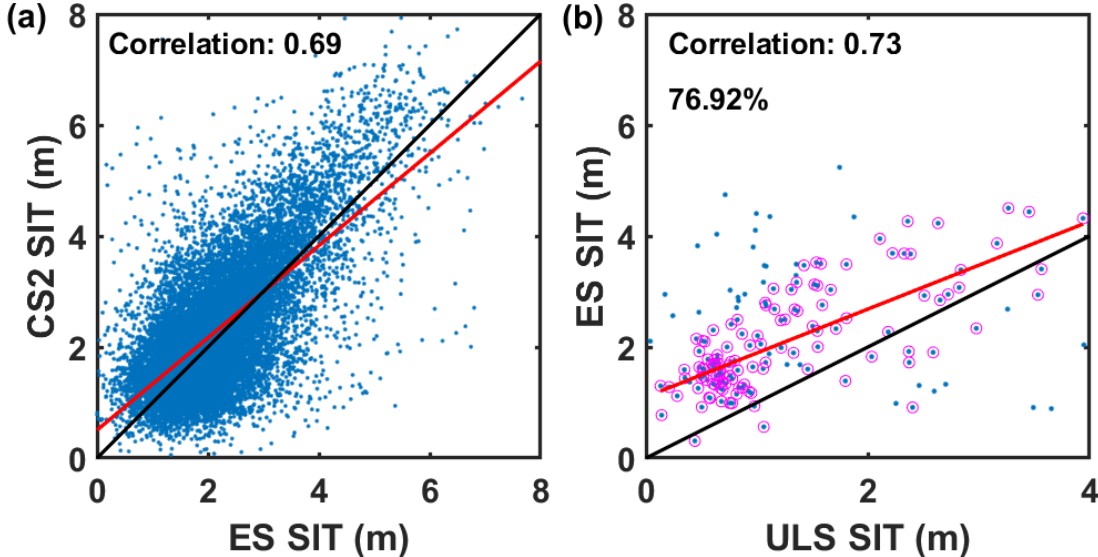

**Figure 2.** (a) The monthly ES and CS2-derived SIT in the Weddell Sea during the coincident segment from November 2010 to November 2011 and (b) the monthly ES-derived SIT and ULS observations during the coincident segments at sites 206, 207, 208, 229, 231 and 233. The red lines are linear regression lines and the black lines are one-to-one lines. The dots surrounded by red circles indicate the ULS SIT is within the uncertainty of ES and the percentage in (b) denotes the proportion of such dots. The correlation and regression line in (b) are only for dots surrounded by red circles.

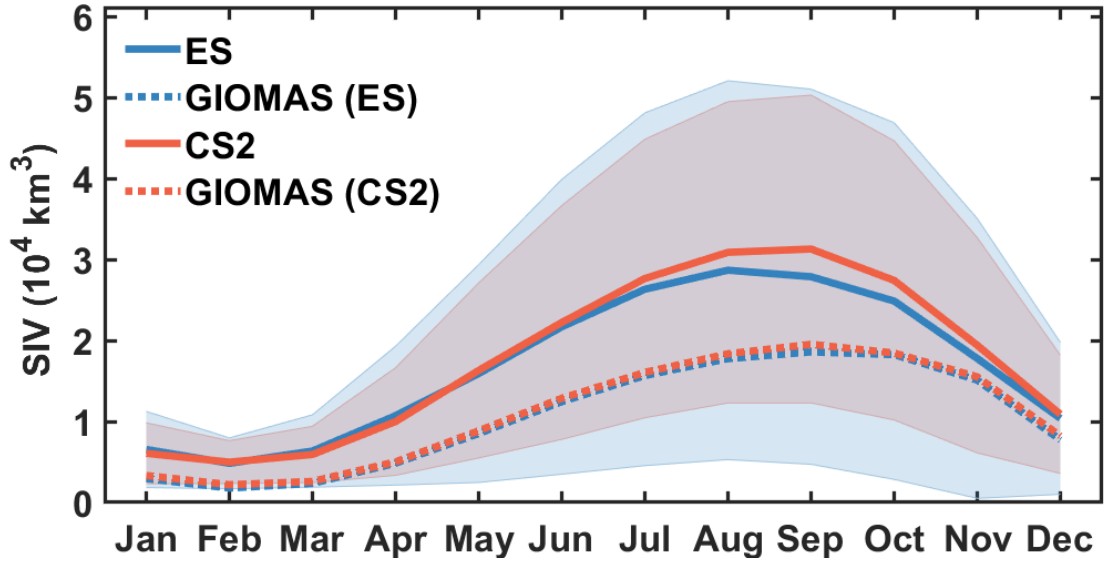


**Figure 3.** The climatological annual cycle of Antarctic SIV. The blue and red denote data related to ES
and CS2, respectively. The solid and dashed curves denote satellite observations and corresponding
GIOMAS data.

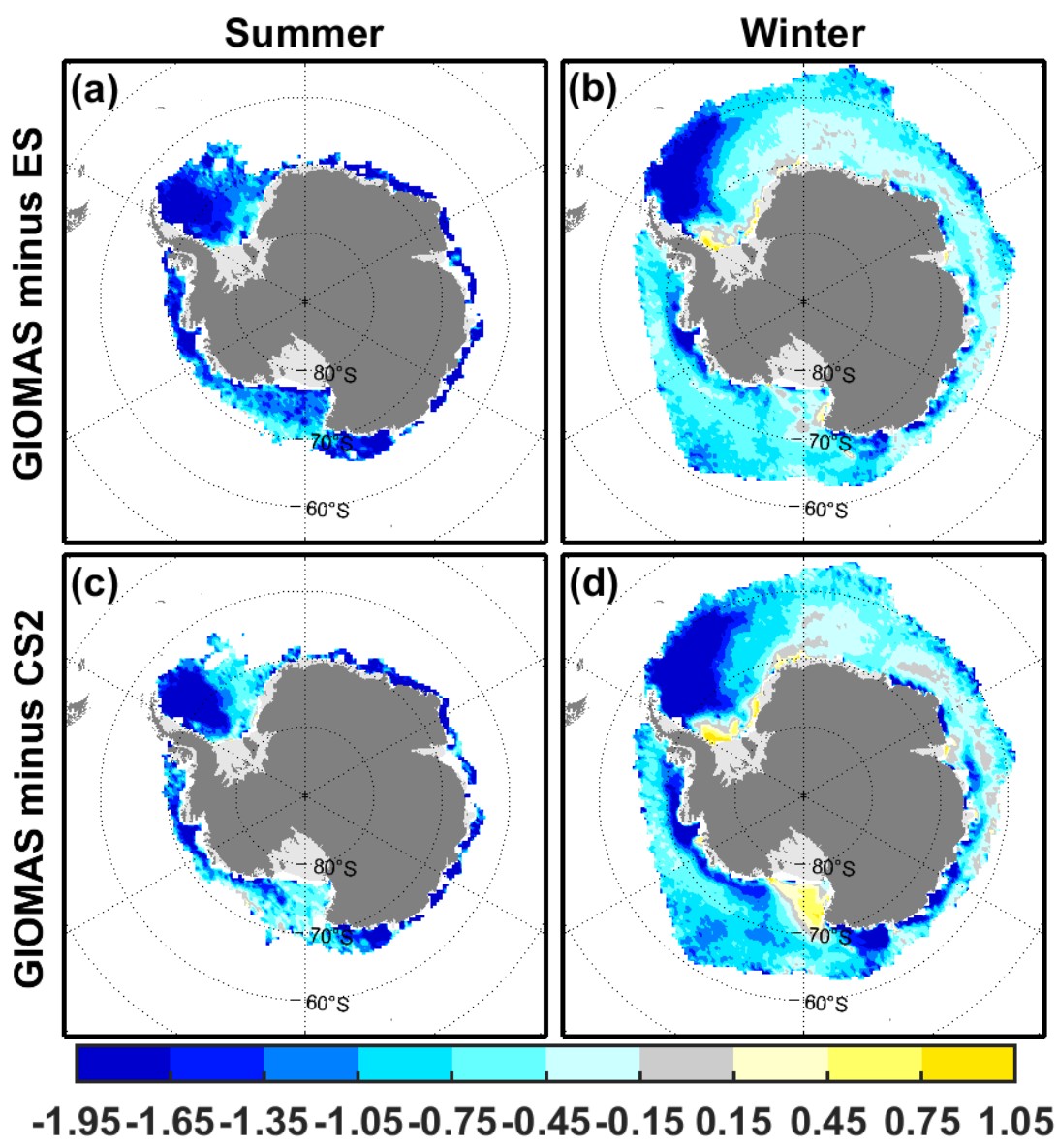


**Figure 4.** The SIT bias of GIOMAS relative to ES in (a) the summer and (b) winter. (c-d) same as (a-b)

but for bias relative to CS2.


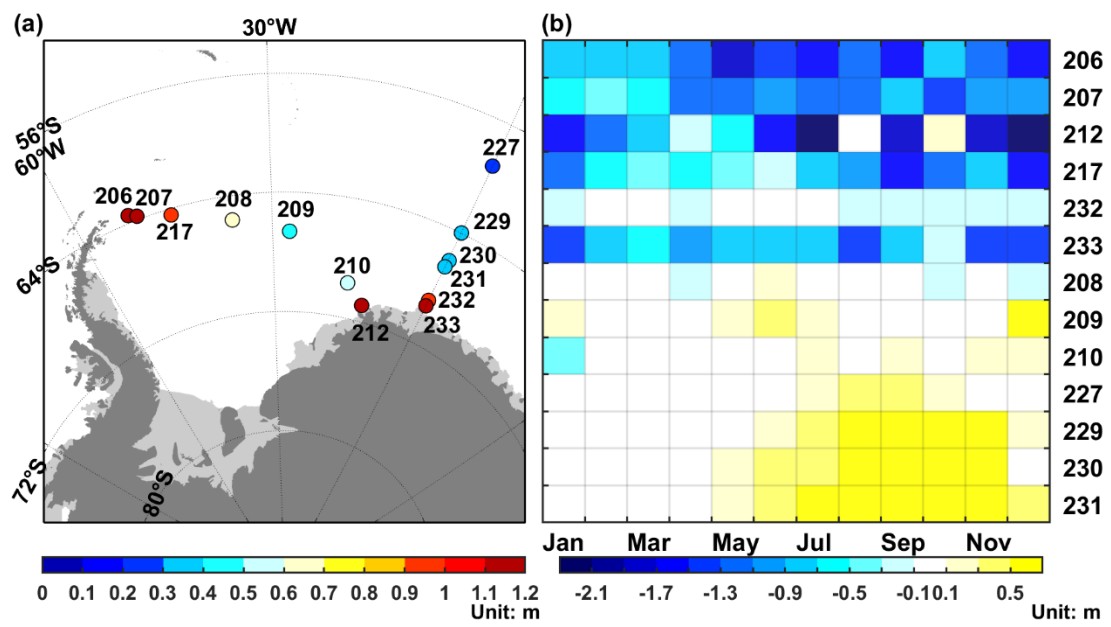

**Figure 5.** (a) The locations of ULS in the Weddell Sea and corresponding standard deviation of SIT. (b) The differences in SIT climatology between GIOMAS and ULS.

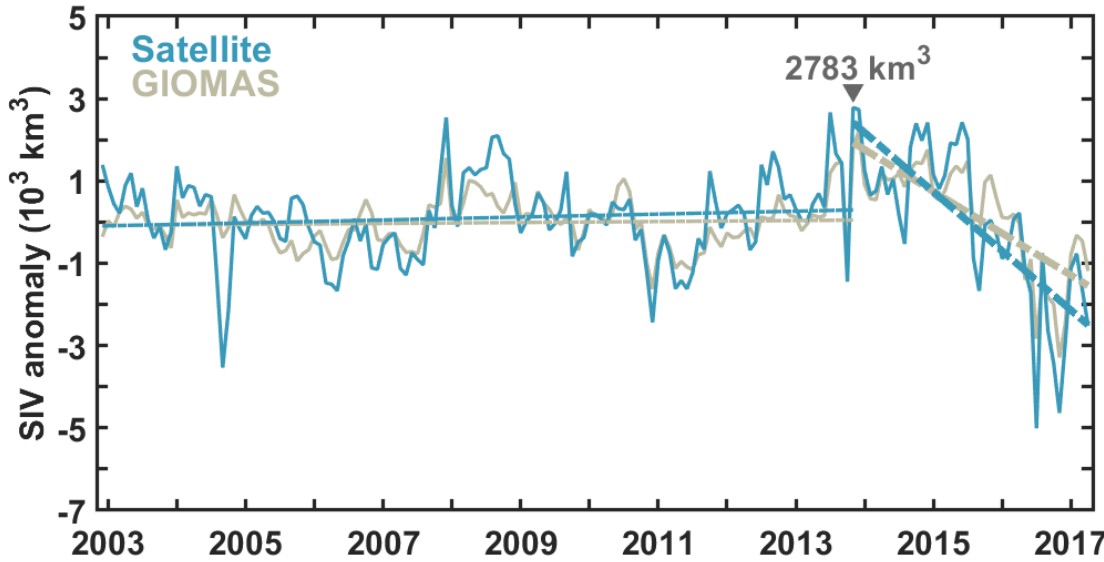


**Figure 6.** The SIV anomalies of satellite observations (green) and corresponding GIOMAS (khaki). The
dashed lines denote the linear trends of SIV anomalies from December 2002 to November 2013 and from
November 2013 to April 2017. All linear trends have passed a F-test at 99% significant level.

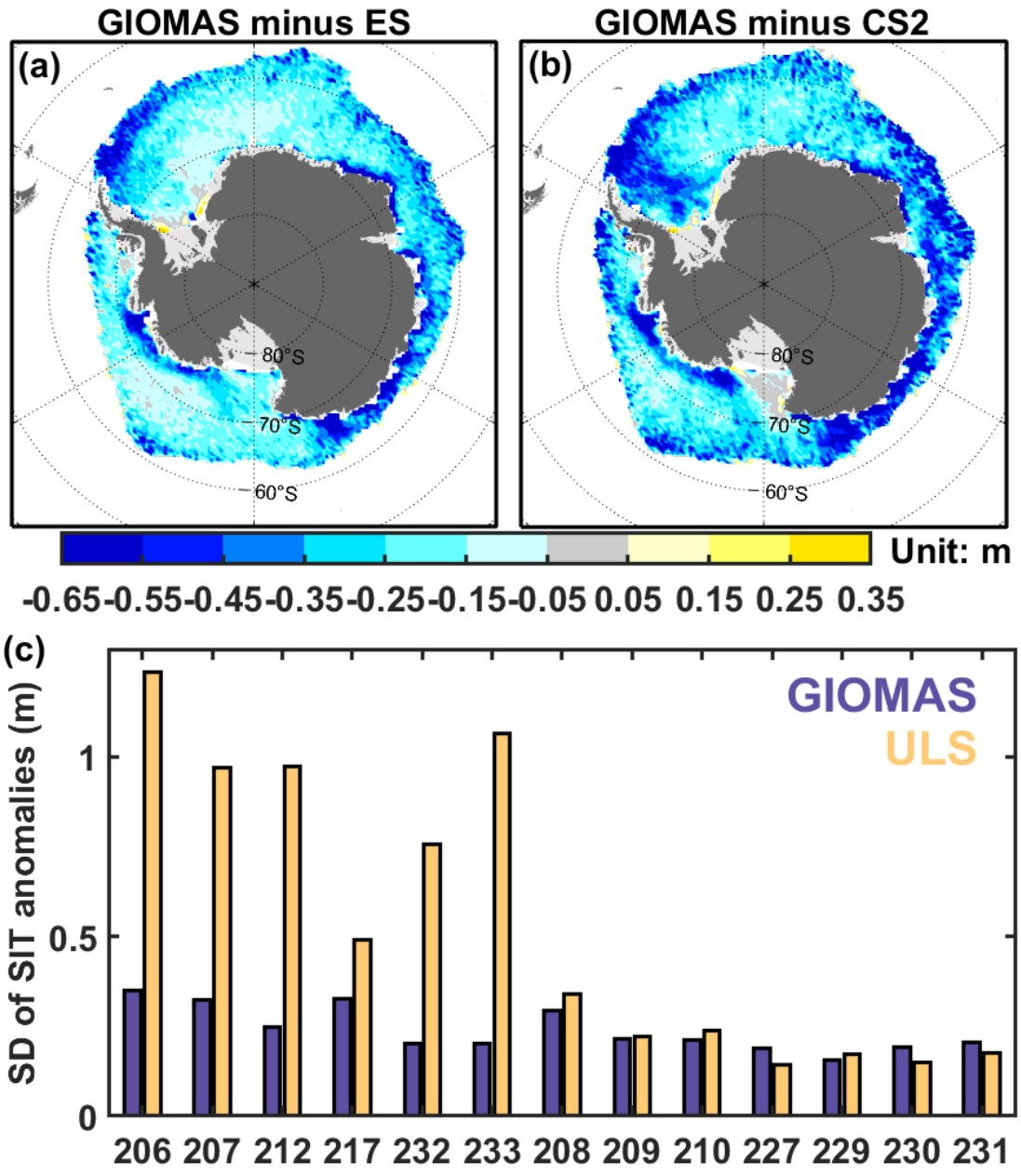

**Figure 7.** (a) The spatial differences in standard deviation of SIT anomalies between GIOMAS and ES. (b) same as (a) but for differences between GIOMAS and CS2. (c) The standard deviation of SIT anomalies for ULS (yellow) and corresponding GIOMAS (blue).

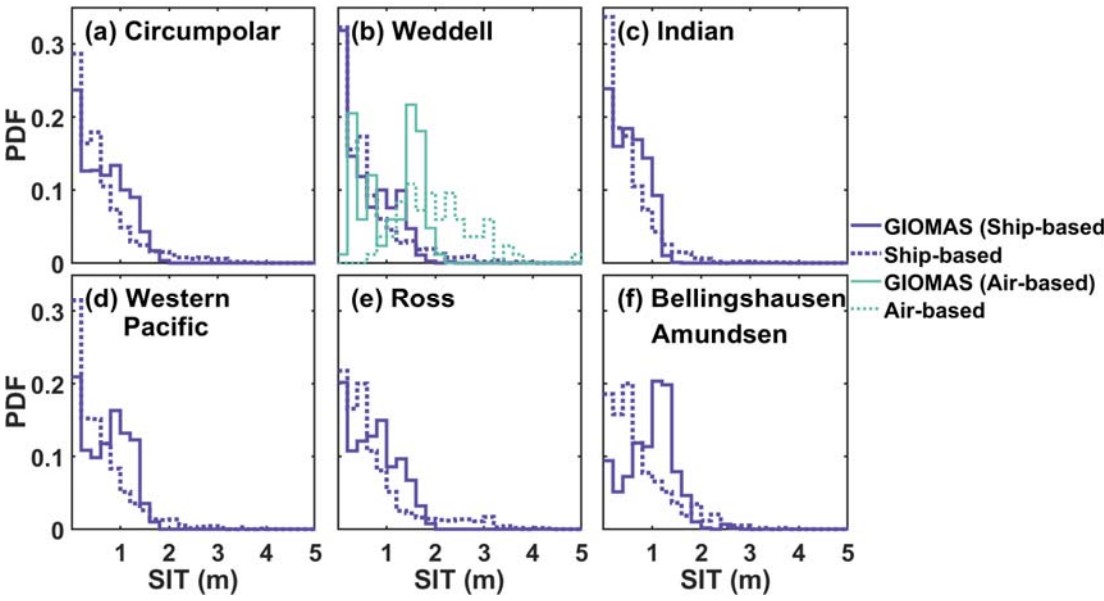

668

669 **Figure 8.** SIT histograms of GIOMAS and in situ observations in (a) the Southern Ocean and (b-f)

670 different sectors.