# Peer review of "An Evaluation of Antarctic Sea-Ice Thickness from the"

_The Cryosphere, 2021_

## Author Comment (AC1)

**Reviewer #1**

Dear Reviewer,

Thank you for your helpful comments and suggestions to improve the manuscript. Various changes are made in the revised manuscript, including but not limited to optimizing the data processing methods, a further analysis of the differences between the reanalysis and observations as well as polishing the language. Below, we repeat each comment and reply to them one by one. All responses are in blue font for clarity of reading.

Hao Luo

On behalf of all the authors

1. Discuss sea-ice thickness in a model run /without/ assimilation of sea-ice concentration. A complementary approach would be to discuss the assimilation increments (here: nudging tendencies) for sea-ice concentration. Assuming that GIOMAS still uses the original PIOMAS nudging methods (Lindsay and Zhang, 2006), there will also be implicit increments on SIT and SIV. It would be good to also discuss the implicit increments on SIT and SIV that arise from SIC the nudging, as their impact on the mean state might be substantial.

*Response:*

Thanks for your suggestion. It is important to clarify the impact of nudging sea-ice concentration (SIC) on the sea-ice thickness (SIT) with your suggested experiments. In Lindsay and Zhang (2006), the "model only" simulation (i.e., model run without data assimilation) and the "Gice-DA" simulation (i.e., model run with nudging SIC) are systematically evaluated to examine the influence of nudging SIC observations, though the experiments are based on the Arctic. The Lindsay and Zhang study shows that the model only case, without being constrained by satellite SIC, generally overestimates SIT in the marginal ice zone of the Arctic. The GIOMAS model only case is likely to also overestimate SIT in the marginal ice zone of the Antarctic. However, it is beyond the scope of this study to have detailed comparison between the GIOMAS runs with and without assimilating SIC observations. In this study, **we mainly focused on detailed assessment of the available GIOMAS Antarctic SIT reanalysis with the constraints of satellite SIC observations, which has not been done before.** The data from the GIOMAS model only case without the constraints of satellite SIC are not standard output and not made available to the public because of the possible uncertainty of SIT in the marginal ice zone.

However, in the revised manuscript, **we added words to point out the likely overestimation of SIT in the marginal ice zone of the Antarctic. We also emphasized the importance of assimilation on the reanalysis and add future work plans to the discussion, which are related to the quantification of the influence of nudging SIC on the SIT in the Antarctic. In addition, we added more explanations on the adjustment of SIT caused by nudging SIC and statements on the impact of this process.**

2. Better quantify observational uncertainty. From the figures presented, it is evident that there are major discrepancies between ES and CS2-derived SIT. The same is likely true when comparing the various in-situ observations with the satellite records - can we have some scatter plots of these please? To clarify, I am not asking for an inter-comparison of observational products - this would clearly be outside the scope of what the authors wanted to present. However, it is necessary to present the model-observation discrepancies in the context of the observational uncertainties, so there needs to be some quantitative discussion of these.

*Response:*

Thanks for your suggestion. To better quantify the uncertainty of satellite observations, the direct comparisons among satellite observations, ULS observations and GIOMAS reanalysis are conducted in the Weddell Sea where ULS SIT observations are available.

The monthly SIT of ES and CS2 during their coincident segment (i.e., November 2010 to November 2011) in the Weddell Sea is displayed in RFig. 1a. **The SIT of ES and CS2 is mainly distributed around the one-to-one line and there is a significant correlation of 0.69 between them, indicating ES-derived SIT is comparable to CS2-derived SIT.** There are also some differences between ES and CS2-derived SIT, which can be due to the physical processes that have not been figured out yet in the retrieval of ES dataset such as the flooding in the complex Antarctic snow stratigraphy as Paul et al. (2018) suggested.

Then the monthly ES SIT is compared with monthly ULS SIT during the coincident segments at sites 206, 207, 208, 229, 231 and 233 (RFig. 1b). CS2 dataset is not involved since there are only four data pairs between CS2 and ULS observations. The distribution of data pairs indicates ES tends to overestimate SIT compared with ULS observations, because the scattering surface of the radar altimeter can be inside the snow (Willatt et al., 2010; Wang et al., 2020). However, 77% of ULS SIT is within the uncertainty of ES (RFig. 1b). Besides, the correlation between the ULS SIT that is within the uncertainty of ES and the corresponding ES SIT is 0.73. **All those indicate ES-derived SIT is comparable to ULS observations when the uncertainty is considered**.

Furthermore, GIOMAS SIT is compared to the monthly ES SIT during the coincident segments between ES and ULS observations at the same sites mentioned above. The root-mean-square errors (RMSEs) at each site are calculated to show the differences between ES and GIOMAS SIT. For all ULS sites involved, the mean RMSE between GIOMAS and ES-derived SIT is 1.66 m, while the mean uncertainty of ES is 1.77 m. Considering the expected large uncertainty of ES owing to the difficulties with the estimation of snow depth and density in the Antarctic (personal communication with Robert Ricker), **the difference between GIOMAS and ES SIT cannot be ignored.**

In the revised manuscript, **we added more discussions about the influence of the uncertainty of satellite data on the evaluation of GIOMAS in the result and conclusion parts.**

[Figure]

RFig. 1 The monthly ES and CS2-derived SIT in the Weddell Sea during the coincident segment from November 2010 to November 2011 (a) and the monthly ES-derived SIT and ULS observations during the coincident segments at sites 206, 207, 208, 229, 231 and 233 (b). The red lines are linear regression lines and the black lines are one-to-one lines. The dots surrounded by red circles indicate the ULS SIT is within the uncertainty of ES and the percentage in (b) denote the proportion of such dots. The correlation and regression line in (b) are only for dots surrounded by red circles.

3. Use at least some of the in-situ observations (the ones who are judged most reliable) as ground truth and verify both satellite observations and GIOMAS against these. This would allow to draw some much-needed conclusions on whether it is the observational data sets or the model (or both) that need to be improved.

*Response:*

    As you suggested, **the ULS observation at site 206 is selected as the ground truth since the ULS SIT at site 206 has relatively long and continuous time series.** The biases of SIT relative to ULS and the uncertainties of each dataset are displayed in RTab. 1.

    The bias of GIOMAS is larger than the uncertainty of ULS while the bias of ES is smaller than the uncertainty of ULS (RTab. 1). **This suggests there is a significant discrepancy between GIOMAS and ULS SIT while ES SIT is comparable to ULS SIT at site 206.**

    Notably, in-situ observations provide relatively accurate estimations in specific points while satellite data provides an average estimation of each grid. Besides, satellite observations could provide relatively long and continuous Antarctic SIT observations with wide spatial coverage. **Therefore, we used various observations to make the evaluation more comprehensive.**

    In the revised manuscript, **we further clarified the limitations of the scarcity of Antarctic observations for our evaluation and the necessity of using various observations in the evaluation.**

RTab. 1 The biases of SIT relative to ULS and the uncertainties of SIT at ULS 206 (Unit: m).

| Dataset | Bias | Uncertainty |
|---------|------|-------------|
| ULS | 0 | 1.17 |
| ES | 0.89 | 1.37 |
| GIOMAS | -1.99 | 0.34 |

4. The discussion of trends (p 6 ll 166 - 175) is very interesting but disappointingly rudimentary. I consider Figure 5 a success for GIOMAS and a central result of the manuscript: it convincingly shows that year-to-year variability as well as trends in GIOMAS and the satellite record correspond quite well. This is an important message, but it needs to be backed up with more analysis (see minor comments). I would also suggest to dedicate a separate subsection to the trends.

*Response:*

Thanks for pointing this out. **We further analyzed the difference in the trend of sea-ice volume (SIV) anomalies between GIOMAS and satellite observations as suggested.**

In the revised manuscript, further discussion about the difference in the trends of SIV anomalies was added based on your detailed comments. **We also dedicated the result section into four separate subsections,** which are the comparison in the climatology of SIT/SIV, the comparison of SIV trend, the comparison in the intensity of SIT variability and the comparison of SIT frequency, respectively.

5. Comparing monthly-mean model fields to in-situ observations at a specific point in space and time might introduce substantial errors, because you are not comparing "like with like". Is it true that the authors use monthly-mean fields from GIOMAS? If daily-mean fields are available, these really ought to be used to compare to in-situ data. The mean and variability of sparsely sampled instantaneous in-situ observations might not meaningfully represent the monthly mean and variability over a large-scale region like a model grid cell. Please specify and discuss how exactly you match in-situ observations to model fields and give some justification why you think the direct comparison is not misleading (see e.g. Janjic (2018) for an introduction/overview of the problem).

*Response:*

As you suggested, **the monthly GIOMAS SIT data was replaced with the daily SIT data for data analysis and the data processing method was modified as follows to eliminate the mismatch of spatial and temporal resolutions between the observations and the model data as described in Janjić et al. (2018):**

Firstly, when compared with satellite-based observations, daily GIOMAS data is interpolated to the grid of satellite data using the linear approach and converted to the monthly averages to eliminate the difference in resolution between daily GIOMAS data and monthly satellite observations.

Secondly, for the comparisons between GIOMAS and ULS, we convert 15-minutely ULS data into daily averages for comparison with daily GIOMAS data and

use the nearest neighbor approach to find the GIOMAS grid cells closest to the ULS locations.

Thirdly, for the comparison between GIOMAS and ship-based and air-based observations, since the observations are very dense in space and the temporal resolution is always shorter than one day, we average it into daily and gridded data based on the GIOMAS grid to create a proper dataset that can be used in direct comparison with daily GIOMAS data.

We found that after using daily GIOMAS data, **the updated Figures are similar to the old ones and the main conclusions of the evaluation have not changed though there are some changes in the figures.**

Therefore, **we revised the methodology part and modified the relevant statistics after using daily GIOMAS data in the revised manuscript.**

6. The manuscript would benefit from language editing, as there are various places with small grammatical errors and slightly inappropriate word choices. Nothing serious though. To give two examples: p 1 l 9f: "not very clear" is too colloquial and imprecise. p 2 l 34: "limited by the short of..." needs to be replaced by "limited by the lack of..."

*Response:*

We are sorry for the errors in grammar and vocabulary. We revised the manuscript and correct those errors as suggested.

7. p 3 l 81: nudging of SIC is not state of the art anymore, it can introduce grave problems with SIT (see Tietsche et al. 2012). Please add some discussion on implicit changes to SIT when nudging SIC in your system.

*Response:*

Thanks. Based on the evaluation of the existing GIOMAS data spanning from 1979 to present, it is found that GIOMAS tends to underestimate the Antarctic SIT compared with observations. In addition to the deficiency of the model, **the assimilation method used in GIOMAS is an important source of errors to introduce the above-mentioned underestimation since SIT was adjusted asymmetrically during the assimilation process as follows:** when SIC is nudged in the system, it will modify the SIT distribution to accommodate the change in SIC, which removes sea ice from the distribution without considering its thickness if modelled SIC is too large, while adds sea ice to the 0.1 m thickness bin if modelled SIC is too small (Lindsay and Zhang, 2006). As shown in Lindsay and Zhang, the nudging of SIC observations tends to lower SIT in the marginal ice zone when compared with the case without nudging.

In the revised manuscript, **we added more descriptions of the assimilation process in GIOMAS to the data part and clarified the influence of nudging SIC on SIT**.

8. p 4 l 115 "grid of observations": in-situ data are not gridded. Please rephrase, e.g. "... take place in observation space, which means that GIOMAS was converted to

the locations of the observations"

*Response:*

We are sorry for this mistake. In the revised manuscript, the sentence was rephrased below as suggested: "GIOMAS data is converted to the locations of the observations when compared with satellite and ULS observations".

9.  p 4 l 119: You claim the difference ES - CS2 is "much less" than the variability, but 574 is almost half of 967! Can you please rephrase to acknowledge that there is considerable differences in CS2-ES SIV? I would like to see a figure with the time series of SIV anomalies as well.

*Response:*

Thanks for your comment. The SIV anomalies of ES and CS2 are displayed in RFig. 2. The coincident segment is from November 2010 to November 2011. The trends before 2013 are 2016.7 $km^3$ per month when the coincident segment is ES data and 2039.2 $km^3$ per month when the coincident segment is CS2 data. It indicates though the SIV anomalies of the two satellites in the overlapped period are different, the difference has little effect on the trend of SIV anomalies before 2013.

Thus, **we rephrased the sentence to acknowledge the difference in SIV anomalies between ES and CS2 in the coincident segment and clarified the difference has little effect on the trend computation in the revised manuscript.**

[Figure]

RFig. 2 The SIV anomalies of ES (blue dotted line) and CS2 (green dotted line).

10. p 4 l 120: Please be more precise on your method to "splice together" the two data sets.

*Response:*

Thanks for your suggestion. In the trend computation, the SIV anomalies of CS2 are chosen during the overlapped time since the selection of data between ES and CS2 has little effect on the trend computation as verified above. ES from December 2002 to October 2010 and CS2 from November 2010 to April 2017 are combined to obtain a

relatively long and continuous SIV time series for trend computation.

**We added more precise descriptions on the calculation of the observed SIV trend in the revised manuscript.**

11. p 5 ll 137-150: The problem with the manuscript that I describe above in major comment (1) is most evident in this paragraph. There is lots of speculation of where discrepancies may come from, but no further insight offered at all. Look at these phrases: "... may be due to model bias...", "... maybe caused by smaller freeboard of CS2 than ES2...", "... still disputed whether radar altimeter signals originate from the snow/ice or snow/air interface...". Without at least an attempt to decide whether discrepancies are due to deficiencies in GIOMAS or the satellite observations, it is difficult to draw any conclusion on where improvements are needed. Maybe it makes sense to treat (at least some of) the in-situ observations as the ground truth, and verify both model and satellite observations against them?

*Response:*

As you suggested, **the ULS observation at site 206 with a relatively long and continuous SIT series is selected as the ground truth.** The monthly SIT time series of GIOMAS, ES and ULS are displayed in RFig. 3.

The results indicate ES SIT is comparable to ULS SIT since 95% of ULS SIT time series are within the uncertainty of ES SIT (RFig. 3). The difference between GIOMAS and ES is larger than the uncertainty of ES (i.e., the RMSE is 3.19 m and the uncertainty is 2.04 m), indicating the difference between GIOMAS and ES SIT cannot be ignored.

Notably, the uncertainty of satellite observations is mainly related to the complex snow condition in the Antarctic that we lack knowledge about (Alexandrov et al., 2010). **Owing to that unresolved issue, it is hard to determine where the differences are from.** Therefore, we give some speculation of where the differences between GIOMAS and satellite observations come from.

In the revised manuscript, **we modified the speculation of where the differences between GIOMAS and satellite observations come from in the deformed sea-ice regions based on the comparison and added discussions about the unresolved problems in satellite observations that affect the conclusions in the regions with undeformed ice.**

[Figure]

RFig. 3 The monthly SIT series of ULS (yellow) at sites 206 from April 2008 to

December 2010 and corresponding ES SIT (blue) and GIOMAS SIT (purple). The error bar means the uncertainty of ES SIT.

12. p 6 l 166ff: Can you please show a figure of Antarctic SIE, to clarify whether SIV anomalies and trends are mostly explained by SIE anomalies and trends, or there is some independence? A scatter plot with SIE/SIV anomalies would also be helpful.

*Response:*

Thanks. The scatter plot with monthly Antarctic SIE/SIV anomalies of ES in the Bellingshausen and Amundsen Seas during 2005-2010 is displayed in RFig. 4a. It suggests the SIV and SIE anomalies in that period are basically not related since the correlation between SIE and SIV anomalies is 0.0628 and not significant. Meanwhile, the time series of SIE and SIV anomalies at that period show a pair of opposite trends (RFig. 4b). **It is concluded that the variation of SIV can be significantly different from that of SIE anomalies at least in some regions and during a certain period of time.** Therefore, it is necessary to study the changes of SIV anomalies in the Antarctic.

In the revised manuscript, **we rephrased the sentence to emphasize the necessity of studying SIV in the Antarctic.**

[Figure]

RFig. 4 The monthly SIV and SIE anomalies of ES (a) and the trends of SIV and SIE anomalies (b) in the Amundsen and Bellingshausen Seas during November 2005 to

December 2010. The red line in (a) is linear regression line.

13. p 6 l 171 and Figure 5: The trend lines after 2013 are quite different. Can you please give the numbers for the trends in GIOMAS and the satellite record, and add to the discussion that GIOMAS seems to underestimate the trend seen in the satellite record?

*Response:*

Thanks a lot for the suggestion. **The trends of SIV anomalies of GIOMAS and satellite observations are 681 and 2039 km³ per month before 2013 and -81233 and -102449 km³ per month after 2013.** It reveals GIOMAS tends to underestimate both the upward trend before 2013 and the downward trend after 2013 observed by satellite.

To figure out what the underestimation in the trends of SIV anomalies may be related to, **the monthly sea surface temperature (SST) of GIOMAS is compared with the Microwave Optimally Interpolated (OI) SST from the Remote Sensing Systems.** The time series of bias in SST anomalies of GIOMAS relative to OI SST in the range of 65°S-40°S (there is basically no sea ice in that region all year round) during 2013-2018 are displayed in RFig. 5. In general, the bias of GIOMAS SST is positive before 2014 (i.e., mean bias before 2014 is 0.02 °C) and negative after 2014 (i.e., mean bias after 2014 is -0.05 °C), which is roughly related to the underestimation in the positive SIV trend before November 2013 and negative SIV trend after November 2013 since higher SST would slow down the increase of SIV while lower SST would slow down the decrease of SIV. **The result indicates there can be a relationship between the differences in the trends of SIV anomalies and the differences in SST between GIOMAS and observations.** However, this relationship needs further verification and quantification and further analysis is added to our future work plan.

In the revised manuscript, **we gave the numbers for the trends of SIV anomalies and added the primary analysis about the relationship between the difference in SST and the difference in the trend of SIV anomalies between GIOMAS and observations to the discussion.**

[Figure]

RFig. 5 The biases (GIOMAS minus observations) of monthly SST anomalies between

GIOMAS and Microwave OISST observations from the Remote Sensing Systems in the range of 65°S-40°S during 2003-2018. The biases are processed using a 12-month running average method to eliminate the seasonal signals. The black line represents the bias is 0.

14. p 6 l 176f: The variability of SIT anomalies in satellite observations quite probably have a contribution from measurement noise, which the model should not try to imitate/simulate. Can you offer some comment or quantification on that point?

*Response:*

   Thanks for your comment. We agree that the uncertainty of satellite observations would influence the conclusion of the evaluation in the variability of SIT anomalies especially in the regions dominated by undeformed sea ice. However, in the highly deformed sea-ice regions such as ULS 206, the difference between GIOMAS and satellite observations is significant (i.e., the RMSE between GIOMAS and ES SIT at ULS 206 is 3.19 m while the uncertainty is 2.04 m). Meanwhile, since the uncertainty of satellite observations is expected to be large due to the difficulties with the estimation of snow depth and density in the Antarctic (personal communication with Robert Ricker), the difference between GIOMAS and satellite observations cannot be ignored even if it is smaller than the uncertainty in some regions.

   In the revised manuscript, **we added more discussions about the influence of the measurement noise of satellite observations on the difference in the variability of SIT between GIOMAS and satellite.**

15. p 6 l 180 ("... which is consistent with Fig. 4b."): This looks like a mistake. Figure 4b shows difference in mean not difference in variability. Please clarify and correct.

*Response:*

   We are sorry about that mistake. In the revised manuscript, we rephrased the sentence as follows: "The spatial distribution of differences in variability is roughly consistent with that of the SIT differences in Fig. 4b".

16. p 6 l 181: I agree there appears to be a relationship to some extent, but the authors might be over-simplifying this relationship. Look at ULS 232: GIOMAS underestimates variability by a factor of 3, but has no bias. To clarify this, can we please have a scatter plot of standard deviation ratio GIOMAS/satellite versus mean bias?

*Response:*

   Thanks for your comment. It needs to be stated that GIOMAS underestimates the climatology of SIT at ULS 232 though the bias is smaller than those at other sites in the deformed sea-ice regions. To further verify this relationship, a scatter plot with standard deviation ratio/SIT bias is provided in RFig. 6 as suggested. The standard deviation ratio of SIT is computed from GIOMAS/satellite and the bias of SIT is computed from GIOMAS minus satellite during their coincident segment (i.e., 2002-2011 for ES and 2010-2017 for CS2). The proportions of the dots with negative bias and ratio within 0-1 are 69% for GIOMAS/ES (RFig. 6a) and 82% for GIOMAS/CS2 (RFig. 6b). **It**

**implies that in most cases, with a negative bias, GIOMAS tends to underestimate the variability of SIT**.

Thus, **we provided RFig. 6 as a supplement for the manuscript to further verify this relationship, added some descriptions for the supplement and revised the conclusion related to the relationship between the bias and variability of SIT in the revised manuscript.**

[Figure]

**RFig. 6** The scatter plot with standard deviations ratio/SIT bias for GIOMAS/ES (a) and GIOMAS/CS2 (b). The standard deviation ratio of SIT is computed from GIOMAS/satellite observations and the bias of SIT is computed from GIOMAS minus satellite observations. The standard deviations and the biases are calculated based on the SIT time series of GIOMAS and satellite observations in each grid cell. Red solid lines are linear regression lines and the black solid lines mean the ratio is 1. The percentages denote the proportion of the dots with negative bias and ratio less than 1.

17. end of section 3: I am not clear what the reader should take away from the frequency comparison. Can you please add a paragraph with the conclusions?

*Response:*

Thanks a lot for pointing this out. Based on the frequency comparison between GIOMAS and ship-based observations, GIOMAS tends to overestimate SIT between 0.6-1.8 m in the Southern Ocean under the premise that ship-based observations always bias low (Timmermann et al., 2004; Williams et al., 2015). Besides, the comparison between GIOMAS and air-based observations further verifies the weakness of GIOMAS in the simulation of sea-ice deformation.

In the revised manuscript, **we added a conclusion to the frequency comparison as described above.**

**Reference:**

Alexandrov, V., S. Sandven, J. Wahlin, and O. M. Johannessen, 2010: The relation between sea ice thickness and freeboard in the Arctic. *The Cryosphere*, **4**, 373-

380.

Janjić, T., and Coauthors, 2018: On the representation error in data assimilation. *Quarterly Journal of the Royal Meteorological Society*, **144**, 1257-1278.

Lindsay, R. W., and J. Zhang, 2006: Assimilation of Ice Concentration in an Ice–Ocean Model. *Journal of Atmospheric and Oceanic Technology*, **23**, 742-749.

Paul, S., S. Hendricks, R. Ricker, S. Kern, and E. Rinne, 2018: Empirical parametrization of Envisat freeboard retrieval of Arctic and Antarctic sea ice based on CryoSat-2: progress in the ESA Climate Change Initiative. *The Cryosphere*, **12**, 2437-2460.

Timmermann, R., A. Worby, H. Goosse, and T. Fichefet, 2004: Utilizing the ASPeCt sea ice thickness data set to evaluate a global coupled sea ice–ocean model. *Journal of Geophysical Research-Oceans*, **109**.

Wang, J., C. Min, R. Ricker, Q. Yang, Q. Shi, B. Han, and S. Hendricks, 2020: A comparison between Envisat and ICESat sea ice thickness in the Antarctic. *The Cryosphere Discussions*, **2020**, 1-26.

Willatt, R. C., K. A. Giles, S. W. Laxon, L. Stone-Drake, and A. P. Worby, 2010: Field Investigations of Ku-Band Radar Penetration Into Snow Cover on Antarctic Sea Ice. *IEEE Transactions on Geoscience and Remote Sensing*, **48**, 365-372.

Williams, G., T. Maksym, J. Wilkinson, C. Kunz, C. Murphy, P. Kimball, and H. Singh, 2015: Thick and deformed Antarctic sea ice mapped with autonomous underwater vehicles. *Nature Geoscience*, **8**, 61-67.

---

## Author Comment (AC2)

**Reviewer #2**

Dear Reviewer,

   Many thanks for your help to improve our work significantly. We made various changes in response to your constructive comments and suggestions, including but not limited to optimizing the data processing methods, a further analysis of the differences between the reanalysis and observations as well as polishing the language. Below, we repeat each comment and reply to them one by one. All responses are in blue font for clarity of reading.

Hao Luo

On behalf of all the authors

1. How do the authors "convert the GIOMAS data to the observed grid"? Do the authors use a nearest neighbor approach? How do you handle multiple entries in case of large resolution differences? Mean? Median? Min? Max? Please elaborate!

*Response:*

   Thanks for your suggestion. **The data processing method is further stated as follows:**

   Firstly, when compared with satellite-based observations, daily GIOMAS data is interpolated to the grid of satellite data using the linear approach and converted to monthly averages to eliminate the differences in resolution between daily GIOMAS data and monthly satellite observations.

   Secondly, for the comparisons between GIOMAS and ULS observations, we convert 15-minutely ULS data into daily averages for comparison with daily GIOMAS data and use the nearest neighbor approach to find the GIOMAS grid cells closest to the ULS locations.

   Thirdly, for the comparison between GIOMAS and ship-based and air-based observations, since the observations are very dense in space and the temporal resolution is always shorter than one day, we average them into daily and gridded data based on the GIOMAS grid to create proper datasets that can be used in direct comparison with daily GIOMAS data.

   In the revised manuscript, **we revised the data processing method and added more details about the method as described above.**

2. How exactly do the authors define the "climatological annual cycle"? From all years available in GIOMAS? Following some standard reference period like, e.g., 1970-2000? Please elaborate. Additionally, do the authors only use the averages or also daily GIOMAS data for their comparisons?

*Response:*

   Thanks a lot for pointing this out. The "climatological annual cycle" is defined as the multi-year averages in each month. For observations, the climatological annual cycles are calculated from all years available in each observation dataset. For GIOMAS, when compared with satellite observations, GIOMAS data that coincides with the time

spans of satellite observations are selected (2002-2011 for ES and 2010-2017 for CS2) to calculate the climatology. When compared with ULS observations, all years available in GIOMAS (1979-2018) are used for the computation of climatology. Besides, daily GIOMAS data is used for the comparisons.

In the revised manuscript, **we added the definition of the climatology to the method part. We also declared daily GIOMAS data is used in the evaluation.**

3. What exactly do the authors mean by "spliced together"? This definitely needs clarification!

*Response:*

Thanks for your comment. We combined the sea-ice volume (SIV) anomalies of ES and CS2 in the time dimension. In the computation of SIV trends, the SIV anomalies of CS2 during the coincident period (November 2010 to November 2011) are chosen since no matter which dataset is selected, the trend changes little (i.e., the trend before 2013 is 2016.7 $km^3$ per month when choosing ES data and 2039.6 $km^3$ per month when choosing CS2 data in the coincident segment). Thus, the SIV anomalies of ES from December 2002 to October 2010 and CS2 from November 2010 to April 2017 are combined to obtain a relatively long and continuous time series for trend computation.

In the revised manuscript, **we revised the paragraph and added more precise descriptions to the computation of the trends of SIV anomalies.**

4. L130-L136 and L145-L150: I disagree with some of the authors suggestions. For example, while the resolution, i.e. the spacing between footprints I assume the authors mean(?), and the footprint size and shape differ substantially between both sensors, the used retracker algorithms are intentionally the same. Following your mentioned reference of Paul et al. (2018), the effort of the SICCI project was to keep things a consistent as possible between sensors to also allow for an as consistent as possible time series.

*Response:*

We are so sorry for this mistake. As Paul et al. (2018) indicated, the mismatch between the ES and CS2-derived sea-ice thickness (SIT) is likely owing to the physical processes that are unresolved such as the flooding in the snow/ice interface.

In the revised manuscript, **we revised the reason why ES dataset differs from CS2 dataset as Paul et al. (2018) indicated**.

5. Furthermore, following the authors own illustrations between Envisat/Cryosat and GIOMAS, the made statements following Schwegmann et al. (2016) are just not applicable anymore. Both references, Paul et al. and Schwegmann et al. refer to different versions of the SICCI data and can to my understanding not be compared. I strongly urge the author to reiterate this paragraph. This also relates to L145/146

*Response:*

We are sorry for this mistake again. In the revised manuscript, **we reiterated this paragraph and revised the reasons for the differences between ES and CS2 observations according to Paul et al. (2018).**

6.  In L146, I disagree with the fact that this is still disputed. I think there is a clear understanding in the community that the scattering surface is NOT the snow/ice interface but everything in the snow is too complex to be able to say WHERE it scatters

*Response:*

Agreed. we rephrased the sentence in the revised manuscript as follows: "some of the radar altimeter signals would originate from the snow/air interface or from somewhere inside the snow and result in an overestimation of ice draft".

7.  A last general point that I find clearly missing in the authors study with all the given data at hand is a proper comparison between all observations, i.e. how do satellite observations and GIOMAS perform directly compared to the observational data? Using not averages but the daily or monthly data. While it is nice to assume the satellite data is correct. As stated above, this is likely not true and an overestimation of reality. A proper analysis of this could really benefit the manuscript and would be of great use to the scientific community!

*Response:*

Thanks for your suggestion. The inter-comparison among observations and GIOMAS reanalysis can indeed help us figure out where the differences between GIOMAS and satellite observations may come from. Thus, **the ULS observation at site 206 with a relatively long and continuous SIT time series is selected as the ground truth and used to verify the corresponding satellite observations and GIOMAS reanalysis (RFig. 1).**

**Result reveals that ES SIT is comparable to that of ULS since 95% of ULS SIT is within the uncertainty of ES.** The biases of ES and GIOMAS relative to ULS are 0.89 m and -1.99 m, respectively, while the standard deviations are 0.34 m for GIOMAS, 1.37 m for ES and 1.17 m for ULS. **Those suggest ES SIT is closer to the ground truth and the difference between GIOMAS and ULS is significant**. Notably, the inter-comparison between observations is not involved in our evaluation because such comparison in the Weddell Sea has been done in previous studies (e.g., Kern et al., 2015; Shi et al., 2021).

In the revised manuscript, **we added more discussions about the influence of the uncertainty of satellite observations on the evaluation of GIOMAS.**

[Figure]

**RFig. 1** The monthly SIT series of ULS at site 206 from April 2008 to December 2010 (yellow) and corresponding ES (blue) and GIOMAS (purple) at ULS 206. The error bar means the uncertainty of ES SIT.

8. As a non-native English speaker (and writer), I still find some of the phrasing and choice of words unintuitive and I recommend some language editing. Example comprise the lack of several "the"'s, the potential use of correct hyphenation (e.g., sea-ice thickness), as well as multiple sentences starting with "And".
   e.g., L18: "of _the_ model"

*Response:*

    We are sorry for those mistakes. We revised the manuscript and polish the language as suggested.

9. L35: "lack" instead of "short"? or "shortage"?

*Response:*

    Agreed. We changed "short" to "lack" as suggested.

10. L35: What do the authors mean with "freshwater flux of the SO?" by means of melting sea ice?

*Response:*

    We mean the freshwater flux of the Southern Ocean partially comes from the sea ice melting and growth processes. **The sentence was rephrased for clearer description in the revised manuscript.**

11. L38/39: This sounds like a pretty exhaustive list- what other data sets would there be? The way the sentence is phrased suggests this is very limited. While of course the _amount_ of data from these sources might be – its not by the number different available sources if the authors know what I mean.

*Response:*

    **Follow your suggestion, the sentence was rephrased in the revised manuscript as follows:** "So far, the commonly used types of the Antarctic SIT data are observations, model data, and reanalysis products and each type of data has its own limitations".

12. L42: The sentence starting with "while" reads clunky

*Response:*

Agreed. We rephrased the sentence as follows: "It is well known that satellite observations have wider spatiotemporal coverage than in-situ observations".

13. L49: I think the plural from reanalysis is reanalyses.

*Response:*

Agreed. We revised the manuscript and change all the plural from "reanalysis" to "reanalyses" as suggested.

14. L50: Stop sentence after "alone." and start the next one with "Reanalyses merge the information […]"

*Response:*

We rephrased the sentence as suggested.

15. L53: Do not start the sentence with "And".

*Response:*

We changed "And" to "Besides".

16. L63: I think it should be algorithmS.

*Response:*

Agreed. We modified as suggested in the revised manuscript.

17. L67: see comment to Line 53.

*Response:*

We changed "And" to "In addition" as suggested.

18. L92: CCI is not properly introduced as abbreviation (only in the context of SICCI)

*Response:*

Agreed. We added the full name of ESA CCI in the revised manuscript as follows: "European Space Agency Climate Change Initiative (ESA CCI)".

19. L93: The authors should probably introduce freeboard a bit earlier already.

*Response:*

Thanks for your suggestion. We rechecked the text and introduced freeboard earlier in the revised manuscript.

20. L108/109: This means an EM Bird like instrument?

*Response:*

Yes, it is. The airborne electromagnetic system is an EM bird like electromagnetic thickness sensor carried by a helicopter as described in Lemke (2014). In the revised manuscript, **we added a simple description for the instrument**.

21. L120: THE trend!

*Response:*

   We modified as suggested in the manuscript.

22. L158: Maybe this is a result from the coarse resolution of the assimilated SSM/I SIC data? Clearly, this would be unable to resolve especially polynya, fast-ice and thin-ice signatures close to the coast.

*Response:*

   Agreed. That reason was added to explain the deficiency of GIOMAS in resolving sea ice near the coast as suggested in the revised manuscript.

23. Figure 1: As a figure enthusiast, I would urge the authors to use a better fitting land/ice-shelf mask for their figures. While the Ronne/Filcher areas are shown as "land" the Ross ice shelf is not visible at all. The authors could consider using data from Natural Earth or other similar sources.

*Response:*

   Thanks for your suggestion. As shown in RFig. 2 (Fig. 1b in the revised manuscript), the better land/ice-shelf mask from Natural Earth was used in Fig. 1b as you suggested. It shows the Ronner/Filcher areas in the Weddell Sea and the Ross ice shelf are well distinguished from the land. The same mask was also used in Figs. 3, 4a, 6a and 6b in the revised manuscript.

[Figure]

**RFig. 2 The spatial coverage of data used in this study.**

24. Figure 2: It might be my printer, but the two different shade colors for the uncertainty are hard to differentiate. I would assume the wider ones belongs to Envisat but I cannot tell for sure.

*Response:*

We are sorry for this problem. As shown in RFig. 3 (Fig. 2 in the revised manuscript), the new color scheme was used to better distinguish the two shades.

[Figure]

**RFig. 3 The climatological annual cycle of Antarctic SIV. The blue and red denote data related to ES and CS2, respectively. The solid and dashed curves denote satellite observations and corresponding GIOMAS data. The shading denotes the SIV uncertainty of satellite observations.**

**References:**

Kern, S., and Coauthors, 2015: The impact of snow depth, snow density and ice density on sea ice thickness retrieval from satellite radar altimetry: results from the ESA-CCI Sea Ice ECV Project Round Robin Exercise. The Cryosphere, **9**, 37-52.

Lemke, P., 2014: The Expedition of the Research Vessel Polarstern to the Antarctic in 2013 (ANT-XXIX/6). 21, Number of 1-154pp.

Paul, S., S. Hendricks, R. Ricker, S. Kern, and E. Rinne, 2018: Empirical parametrization of Envisat freeboard retrieval of Arctic and Antarctic sea ice based on CryoSat-2: progress in the ESA Climate Change Initiative. *The Cryosphere*, **12**, 2437-2460.

Schwegmann, S., E. Rinne, R. Ricker, S. Hendricks, and V. Helm, 2016: About the consistency between Envisat and CryoSat-2 radar freeboard retrieval over Antarctic sea ice. *The Cryosphere*, **10**, 1415-1425.

Shi, Q., Q. Yang, L. Mu, J. Wang, F. Massonnet, and M. R. Mazloff, 2021: Evaluation of sea-ice thickness from four reanalyses in the Antarctic Weddell Sea. *The Cryosphere*, **15**, 31-47.

Wang, J., C. Min, R. Ricker, Q. Yang, Q. Shi, B. Han, and S. Hendricks, 2020: A comparison between Envisat and ICESat sea ice thickness in the Antarctic. *The Cryosphere Discussions*, **2020**, 1-26.

Willatt, R. C., K. A. Giles, S. W. Laxon, L. Stone-Drake, and A. P. Worby, 2010: Field Investigations of Ku-Band Radar Penetration Into Snow Cover on Antarctic Sea Ice. *IEEE Transactions on Geoscience and Remote Sensing*, **48**, 365-372.

---

## Author Response (AR2)

**Reviewer #1**

Dear Reviewer,

    Thank you for your helpful comments and suggestions to improve the manuscript. Further changes are made in the revised manuscript, including adding discussions about the observation error estimation used in GIOMAS, necessary figure, and a table together with the explanations and discussion. Below, we repeat each comment and reply to them one by one. All responses are in a blue font for clarity of reading.

Hao Luo

On behalf of all the authors

1. Regarding the inclusion of additional model diagnostics or simulations, I am disappointed that the authors could not include additional analysis - this is a missed opportunity to give the paper much more depth and value. However, I accept the author's decision that this is outside of the scope of their paper and can be subject of future research. The inclusion of additional discussion and caveats in the revised manuscript is appreciated.

*Response:*

    Thanks for your suggestion. **In the revised manuscript, we added more discussions about the observation error estimation used in the GIOMAS as follows**: "Thirdly, due to the characteristics of SIC observations derived from the passive microwave instrument, observation errors of SIC should vary with time and location (Lindsay and Zhang, 2006). However, a fixed value of observation errors is adopted in GIOMAS because of limited information on observation errors of SIC. Thus, a better estimation of SIC observation errors might further improve the performance of GIOMAS." (*Please see lines 387-391 in our revised manuscript*)

2. The new analysis on observational uncertainty in the authors' response is a nice piece of work, however it is not adequately reflected in the revised manuscript. I would insist that Fig 1 and Tab 1 in the response together with the accompanying explanations and discussions are added to the manuscript, to put the model-observation discrepancies in the context of observational uncertainties.

*Response:*

    Thanks for your suggestion. In the revised manuscript, **RFig. 1 and RTab. 1 and related captions were added** (*Please see Figure 2 and Table 1 in our revised manuscript*). **The explanation and discussion of RFig. 1 was added into the introduction of observations in the data part as follows**: "Since both satellite observations and in situ observations are involved in the evaluation, it is necessary to investigate the relationship between them. To achieve this, direct comparisons among satellite observations and ULS observations are conducted in the Weddell Sea where ULS SIT observations are available. The monthly SIT of ES and CS2 during their coincident segments (i.e., November 2010 to November 2011) in the Weddell Sea is displayed in Fig. 2a. The SIT of ES and CS2 is mainly distributed around the one-to-one line and there is a significant correlation of 0.69 between them, indicating ES-

derived SIT is comparable to CS2-derived SIT. Then the monthly ES SIT is compared with monthly ULS SIT during the coincident segments at sites 206, 207, 208, 229, 231 and 233 (Fig. 2b). CS2 dataset is not involved since there are only four data pairs between CS2 and ULS observations. The distribution of data pairs indicates ES tends to overestimate SIT compared with ULS observations, because the scattering surface of the radar altimeter can be inside the snow (Willatt et al., 2010; Wang et al., 2020). However, 77% of ULS SIT is within the uncertainty of ES (Fig. 2b). Besides, the correlation between the ULS SIT that is within the uncertainty of ES and the corresponding ES SIT is 0.73. All those indicate ES-derived SIT is comparable to ULS observations when the uncertainty is considered." (*Please see lines 144-157 in our revised manuscript*)

[Figure]

**RFig. 1** (a) The monthly ES and CS2-derived SIT in the Weddell Sea during the coincident segment from November 2010 to November 2011 and (b) the monthly ES-derived SIT and ULS observations during the coincident segments at sites 206, 207, 208, 229, 231 and 233. The red lines are linear regression lines and the black lines are one-to-one lines. The dots surrounded by red circles indicate the ULS SIT is within the uncertainty of ES and the percentage in (b) denotes the proportion of such dots. The correlation and regression line in (b) are only for dots surrounded by red circles.

Meanwhile, **the explanations and discussion of RTab. 1 was added into the results part as follows**: "To prove this, a direct inter-comparison between the monthly SIT of GIOMAS, ES and ULS at site 206 is conducted and the biases of GIOMAS and ES relative to ULS at site 206 and the uncertainties of each dataset are displayed in Table 1. The bias of GIOMAS is larger than the uncertainty of ULS while the bias of ES is smaller than the uncertainty of ULS (Table 1). This suggests there is a significant discrepancy between GIOMAS and ULS/ES SIT while ES SIT is comparable to ULS SIT at site 206." (*Please see lines 229-234 in our revised manuscript*)

**RTab. 1** The biases of GIOMAS and ES SIT relative to that of ULS and the

uncertainties of SIT at ULS 206 (Unit: m).

| Dataset | Bias | Uncertainty |
|---------|------|-------------|
| ULS | 0 | 1.17 |
| ES | 0.89 | 1.37 |
| GIOMAS | -1.99 | 0.34 |

3. This is a misunderstanding - I was asking for total Antarctic SIE/SIV, not regional. I would expect the correlation between SIE and SIV to be higher for total Antarctic than for the Amundsen and Bellingshausen Seas as shown in Figure 4 of the response. Can you please provide the figures for the entire Antarctic, and modify the text in the manuscript accordingly?

*Response:*

We are sorry for the misunderstanding. As you suggested, the sea-ice extent/sea-ice volume (SIE/SIV) anomalies and the corresponding trend for the whole Antarctic are shown in RFigs. 2a-b. It is suggested that the total Antarctic SIV and SIE anomalies have similar linear trends while the correlation between them is 0.407 and has passed a 99% F test. **Those indicate a relatively good consistency between Antarctic SIE and SIV as you mentioned.**

Due to the distinct regional differences in the variation of Antarctic sea ice (e.g., Parkinson, 2019), the trends of SIV/SIE anomalies in the whole Antarctic and different sectors are shown in RFig. 2c. For the whole Antarctic, the trends are the same as those in RFig. 2a. However, for the Amundsen Sea and the Bellingshausen Sea, the trends of SIV and SIE are opposite. Meanwhile, for the Weddell Sea and the Ross Sea, differences can also be found in the trends of SIV and SIE. This implies that it is necessary to investigate the Antarctic SIV variation.

In the revised manuscript, **the statements were added to clarify the significant correlation between Antarctic SIE and SIV** (*Please see lines 256-257 in our revised manuscript*).

[Figure]

**RFig. 2** (a) The monthly SIV and SIE anomalies and (b) their scatter plot in the Antarctic. (c) The scatter plot of the trends of SIV and SIE anomalies in the whole Antarctic and different sectors. The time of the data is from November 2005 to December 2010. The red line in (b) is a linear regression line and the gray dotted lines in (c) are zero lines.

**Reference:**

Lindsay, R. W., and J. Zhang, 2006: Assimilation of Ice Concentration in an Ice–Ocean Model. *Journal of Atmospheric and Oceanic Technology*, **23**, 742-749.

Parkinson, C. L., 2019: A 40-y record reveals gradual Antarctic sea ice increases followed by decreases at rates far exceeding the rates seen in the Arctic. *Proceedings of the National Academy of Sciences of the United States of America*, **116**, 14414.

Wang, J., C. Min, R. Ricker, Q. Yang, Q. Shi, B. Han, and S. Hendricks, 2020: A comparison between Envisat and ICESat sea ice thickness in the Antarctic. *The Cryosphere Discussions*, **2020**, 1-26.

Willatt, R. C., K. A. Giles, S. W. Laxon, L. Stone-Drake, and A. P. Worby, 2010: Field Investigations of Ku-Band Radar Penetration Into Snow Cover on Antarctic Sea Ice. *IEEE Transactions on Geoscience and Remote Sensing*, **48**, 365-372.